# Preclinical Development of Orally Inhaled Drugs (OIDs)—Are Animal Models Predictive or Shall We Move Towards In Vitro Non-Animal Models?

**DOI:** 10.3390/ani10081259

**Published:** 2020-07-24

**Authors:** Dania Movia, Adriele Prina-Mello

**Affiliations:** 1Laboratory for Biological Characterisation of Advanced Materials (LBCAM), Department of Clinical Medicine, Trinity Translational Medicine Institute, Trinity College, The University of Dublin, Dublin D8, Ireland; prinamea@tcd.ie; 2AMBER Centre, CRANN Institute, Trinity College, The University of Dublin, Dublin D2, Ireland

**Keywords:** respiratory diseases, inhalation, preclinical studies, drug development, non-animal methods

## Abstract

**Simple Summary:**

This commentary focuses on the methods currently available to test the efficacy and safety of new orally inhaled drugs for the treatment of uncurable respiratory diseases, such as chronic obstructive pulmonary disease (COPD), cystic fibrosis or lung cancer, prior to entering human experimentation. The key question that the authors try to address in this manuscript is whether there is value in using and refining current animal models for this pre-clinical testing, or whether these should be relinquished in favor of new, more human-relevant non-animal methods.

**Abstract:**

Respiratory diseases constitute a huge burden in our society, and the global respiratory drug market currently grows at an annual rate between 4% and 6%. Inhalation is the preferred administration method for treating respiratory diseases, as it: (i) delivers the drug directly at the site of action, resulting in a rapid onset; (ii) is painless, thus improving patients’ compliance; and (iii) avoids first-pass metabolism reducing systemic side effects. Inhalation occurs through the mouth, with the drug generally exerting its therapeutic action in the lungs. In the most recent years, orally inhaled drugs (OIDs) have found application also in the treatment of systemic diseases. OIDs development, however, currently suffers of an overall attrition rate of around 70%, meaning that seven out of 10 new drug candidates fail to reach the clinic. Our commentary focuses on the reasons behind the poor OIDs translation into clinical products for the treatment of respiratory and systemic diseases, with particular emphasis on the parameters affecting the predictive value of animal preclinical tests. We then review the current advances in overcoming the limitation of animal animal-based studies through the development and adoption of in vitro, cell-based new approach methodologies (NAMs).

## 1. Introduction

### 1.1. The Current Burden of Respiratory Diseases

Respiratory diseases constitute a huge burden in our society. It has been calculated that, worldwide, around 235 million people are living with asthma [1], 251 million with chronic obstructive pulmonary disease (COPD) [2], and more than 70,000 people with cystic fibrosis [3]. Furthermore, 3 million people are affected by idiopathic pulmonary fibrosis (IPF) [4], and 10 million people contract tuberculosis (TB) annually [5]. In addition to this, lung cancer continues to be the leading cause of cancer death worldwide, accounting for 1.8 million deaths in 2018 [6]; whereas, pneumonia still constitutes the single largest infectious cause of death in children worldwide, with 808,694 deaths under the age of five in 2017 alone, accounting for the 15% of all deaths of children under five years of age [7]. Furthermore, COVID-19 has recently become known across the globe as a respiratory disease with high mortality, particularly in high-risk categories [8]. Consequently, the global respiratory drug market is currently growing at an annual rate ranging from 4% to 6%, depending on the reports, and the leading companies in terms of market share are GlaxoSmithKline, AstraZeneca, Merck, Novartis and Boehringer Ingelheim [9,10]. 

However, respiratory drug development currently suffers of an overall attrition rate of around 70% [9]. It has been calculated that, the cumulative probability to reach the clinical market for drugs targeting respiratory diseases is equal to 3%, compared to the 6–14% probability that applies to drugs used to treat other diseases [11]. The problem is certainly multifactorial and has several contributory causes [12], including, to name a few, general poor understanding of the underlying mechanisms of respiratory diseases, difficulties in drug formulation, and poor performances of the drug administration methods. Nevertheless, it is believed that the limitations of the current preclinical models play a major role in the high attrition rate of respiratory drugs. Our commentary focuses specifically on the preclinical methods currently used in the development of orally inhaled drugs (OIDs), the limitations of these methods, to what degree they affect the translation rate of OIDs into clinical products, and how in vitro, cell-based new approach methodologies (NAMs) could potentially support overcoming the limitations of preclinical methods whilst reducing, or even completely replacing, the need for animal studies.

### 1.2. Inhalation Therapy

Inhalation is the preferred administration method for treating respiratory diseases [13], as: (i) it delivers the drug directly at the site of action, resulting in a rapid therapeutic onset with considerably lower drug doses, (ii) it is painless and minimally invasive thus improving patients’ compliance, and (iii) it avoids first-pass metabolism, providing optimal pharmacokinetic conditions for drug absorption and reducing systemic side effects [14,15,16].

It should be noted here, inhalation differs from intranasal administration for the drug portal-of-entry (PoE) and targeted site of action. Intranasal drugs are sprayed into the nostrils, producing a local effect in the nasal mucosa; whereas, inhalation occurs through the mouth, with the OIDs, also referred to as orally inhaled drug products (OIPs), having their efficacy in the lungs. Notably, attempts have been made to develop OIDs that exert their therapeutic action outside the lung, for the treatment of systemic diseases [17]. The latter include, for example, migraine headaches, treated with aerosols of ergotamine or hydroxyergotamine, and type 1/type 2 diabetes, for which inhaled insulin products have been developed (e.g., Exubera—withdrawn in 2008 due to poor revenue—and Afrezza—the uptake of which has also been impacted by socio-economic issues).

OID therapeutic categories currently approved for the clinical treatment of respiratory diseases include drugs for the treatment of asthma and COPD, such as β2 adrenergic agonists (e.g., albuterol, formoterol) and muscarinic antagonists (e.g., ipratropium, tiotropium) inducing bronchodilation, or glucocorticosteroids (e.g., fluticasone and budesonide) reducing inflammation. OIDs for the treatment of cystic fibrosis are also available for clinical use, with most of them falling into the therapeutic categories of mucolytics (e.g., saline and acetyl choline), aiming at thinning the mucus for facilitating its clearance from the patient’s lungs. Alternatively, leukocyte DNAse, reducing inflammation, and antimicrobial agents (e.g., tobramycin), treating the bacterial infection characteristic of this disease, are also administered as OIDs.

Various devices can be used to administer OIDs to patients, including dry-powder inhalers (DPIs), pressurized metered-dose inhalers (pMDIs) and nebulizers. These devices have been extensively discussed in several recent works [18,19,20,21,22,23,24,25]. Briefly, DPIs deliver powder particles carrying the drug; pMDIs and nebulizers generate liquid droplets containing the drug. To be effective, an inhalation device must be easy to use and forgiving of poor patient’s compliance, while providing reproducible effective dosing. Thus, a through characterization of the performance of the inhalation device is required at regulatory level, when developing an OID. Such characterization is based on in vitro, ex vivo and in vivo (on human volunteers) tests, as extensively described in the scientific literature [26,27,28,29,30,31,32,33] and in the guidelines published by the European Medicine Agency (EMA) (CPMP/EWP/4151/00; CPMP/EWP/239/95; CPMP/180/95; CPMP/QWP/158/96; CPMP/OWP/2845/00; EMEA/CHMP/QWP/49313/2005; CPMP/EWP/4151/00 and EMEA/CHMP/EWP/48501/2008 Appendix 1). Animal models are not used in the characterization of the efficiency and reproducibility of inhalation delivery devices. This is due to the fact that, DPIs and pMDIs are breath-actuated and therefore not compatible with animal exposure; whereas for nebulizers modifications are needed in line with the animal model adopted. Thus, our manuscript, which focuses on the potential reduction and replacement of animals studies in OID development, does not discuss the impact of inhalers’ performance on the effectiveness of inhalation therapies [34], a current challenge discussed in detail elsewhere [35,36,37,38,39,40,41,42,43,44].

Despite the major advantages over i.v. administration of drugs, inhalation therapy encounters several obstacles in achieving an effective therapeutic dose for the successful treatment of respiratory and/or systemic diseases. Below, we describe the journey of an OID once administered and the human-specific features that, in the authors’ opinion, strongly impact on the current low translation rate of OIDs, as these are poorly replicated in the current preclinical models.

#### The Journey of an OID in Patient and Human-Specific Features Impacting on OIDs’ Poor Translation Rate

When an OID is administered to a patient, its liquid or powder aerosol enters the human respiratory system via the oropharynx. OID deposition in the oropharynx is invariably wasteful, reducing the OID dose reaching the lungs. This indeed constitutes the first feature to keep into account for developing an effective inhalation therapy [45]. Rodent models cannot reproduce this feature, as they are obliged nose-breathers. However, other animal models (e.g., dogs) can be used to overcome the limitations posed by rodents. Also, OID deposition in the oropharynx must be minimized in clinics to avoid severe side-effects in the patients. Side-effects can be due to both local and systemic toxicity, as OIDs accumulating in the mouth and throat enter the body through swallowing.

Achieving an optimal OID deposition pattern in the patients’ lung is the second feature to keep into account for an effective inhalation therapy [46]. To reach its site of action and/or absorption, the OID needs to pass through the so-called extrathoracic (or ET) region of the larynx, enter the tracheobronchial region and reach the small and/or peripheral (alveoli) airways. Drug absorption and translocation into the blood flow can in fact occur from all parts of the lung, but it occurs more readily in the alveoli [47], where there is a large surface area and a relatively thin layer of epithelial and endothelial cells separating the inhaled drug from the blood flow. The OID journey within the complex, branched structure of the human lung is influenced by two main parameters of the particles/droplets carrying the drug [48]: (i) velocity [49]; and (ii) aerodynamic size distribution (the so-called APSD) [13]. Both parameters strongly impact on the drug deposition pattern and, subsequently, on the effectiveness of the inhalation therapy. Velocity is defined by the delivery system employed in the OID administration. Generally, high velocity results in increased deposition in the oropharynx and tracheobronchial regions; whereas, low velocity generates a peripheral deposition pattern [13]. It goes without saying that OIDs cannot reach those part of the respiratory tract where velocity is null, i.e., those parts of the lung that are not ventilated. This is particularly relevant to consider when developing OIDs against respiratory diseases [50], which are characterized by the partial or full obstruction of the respiratory tract (e.g., asthma, COPD, cystic fibrosis and lung cancer). Combination of drugs where bronchodilators or mucolytics are used in a synergistic manner with other drug therapies, can be used to modulate OID velocity and increase the efficacy of the inhalation therapy. In parallel, the deposition mechanism of the aerosol particle/droplets in the bronchial tree changes depending on their APSD [13]. Droplets/particles with large aerodynamic size deposit by impaction or interception mechanisms in the oropharynx or just beyond the trachea bifurcation. The smaller droplets/particles deposit in the smaller airways by sedimentation, subject to gravity. Among those, the droplets/particles with aerodynamic size below 3 µm further move to the alveoli by diffusion or Brownian motion. It should be noted here, droplets/particles deposition follows Stokes’ law [45]. The consequence is that, since most of the droplets/particles are near spherical, their aerodynamic size can be small despite being geometrically large. This happens when particles/droplets have low density, which is determined by the composition of the OID formulation. OID deposition pattern is currently evaluated in in vitro, cell-free experiments, achieving good predictive value [51].

Once the OID deposits on the airways, removal mechanisms, such as mucociliary clearance in the conducting airways and macrophage clearance in the alveolar space, can be responsible for the drug elimination and/or degradation [52], thus hindering the local efficacy and/or the systemic absorption of the OID. Mucociliary clearance is the upward movement of mucus driven by beating cilia towards the pharynx, where mucus is subsequently swallowed and pass into the gastrointestinal tract [53]. In macrophage clearance, the OID is phagocytosed by alveolar macrophages and cleared by transport to the lung-draining lymph nodes [54,55]. Compared with mucociliary clearance, macrophage clearance is far slower [56] and, therefore, its action is typically assumed to be negligible for OIDs, unless the drug is known to be degraded by alveolar macrophages [57]. Absorptive drug clearance is yet another clearance mechanism by which an OID is cleared from the lung through the blood circulation, a mechanism that is heavily dependent on perfusion. Perfusion levels, however, vary between the different lung regions. In the alveoli, perfusion levels are the highest and drugs have a very short half-life; by contrast, in the tracheobronchial region, the perfusion rate is lower, thus offering a longer drug bioavailability [58]. Removal mechanisms constitute the third feature to keep into account for developing an effective inhalation therapy. As described in detail in Section 2.1.1, this feature is species-specific [59] and, therefore, human-specific removal mechanisms are not replicated by animal models. Notably, human-specific removal mechanisms can be reproduced by in vitro, cell-based NAMs [60,61,62,63], as discussed in detail in Section 2.2.

To exert local or systemic efficacy, OID dissolution and absorption are indeed necessary [64]. The thickness and constitution of the pulmonary lining fluid, which can be modified by lung diseased states [65], influence OID dissolution and, subsequently, absorption [66], constituting the fourth feature to keep into account for developing an effective inhalation therapy. While the mucus layer (produced by goblet cells in the bronchial region) acts as a physical barrier, surfactants produced by alveolar cells in the peripheral airways reduce surface tension and facilitate drug dissolution [13]. Noteworthy, OID dissolution rates strongly depend on disease-specific airway characteristics (e.g., COPD is characterized by a thick mucus, hindering OID efficacy), which are not replicated by conventional preclinical models. Noteworthy, in vitro, cell-based NAMs have the potential to reproduce the disease-specific composition of pulmonary lining fluid [67].

Finally, the multicellular composition of the lung is the fifth feature to keep into account for developing an effective inhalation therapy, by playing an important role in defining OID delivery efficiency. For example, mast cells have protective functions against inhaled drugs; dendritic cells, together with macrophages, are the first line of defense of the lung immune system, sampling for and removing constantly any exogenous material such as drugs. Clara cells are involved in OID metabolism. Interestingly, the human lung has relatively low metabolic activity as compared to the gastro-intestinal tract or the liver [68,69]. This constitutes a distinct advantage for inhalation therapy over oral drug administration. However, protease activity is generally increased in lung diseases as a result of chronic inflammation (e.g., enhanced activity of cytochrome p450 in patients affected by lung cancer [70,71] or COPD [72]); this can indeed reduce the biopersistence and bioavailability of some OIDs (e.g., insulin [73]). Protection against metabolic activity has been achieved in inhalation therapy by drug encapsulation into carriers (e.g., liposomes [74,75,76,77,78]). Animal models and humans differ in the metabolism and distribution/types of cell populations lining the airways. For example, it has been shown that the average number of cells per alveolus for rats versus humans is: 21 vs. 1,481 for endothelial cells, 13 vs. 106 for interstitial cells, 6 vs. 67 for epithelial type II cells, 4 vs. 40 for epithelial type I cells, and 1.4 vs. 12 for alveolar macrophages [79]. This has important clinical implications during the OID development. Notably, the human-specific composition and metabolism of the lung can indeed be replicated more closely by adopting in vitro, cell-based NAMs, as described in the following sections.

Based on the multiple mechanisms and processes described above, it is evident that OID development is not an easy task. Overall, a sound understanding of the features involved in the OID journey is necessary to use the most predictive preclinical models to overcome the complex, intrinsic challenges associated with inhalation therapy. Interestingly, such challenges have certainly not hindered the interest of the pharmaceutical industry in inhalation therapy. Based on a search carried out by the authors in July 2020, 2542 inhalation clinical trials for new, combination, and existing products, encompassing 666 drug interventions, 1111 different conditions and 115 rare diseases, have been logged on ClinicalTrials.gov in the last four years (search terms: interventional studies; inhalation; start date from 01/01/2016 to 31/12/2020). To put this into context, a total of 97,744 interventional studies, comprising 2867 drug interventions, have been registered on ClinicalTrials.gov in the same time period. Consequently, inhalation clinical trials make for the 2.6% of the total number of interventional studies registered in the time period under consideration (2016–2020), and 23.2% of the total drug interventions examined. It is important to observe that more than half of these inhalation studies are for systemic conditions, thus demonstrating an interest that expands beyond the domain of respiratory diseases.

## 2. Discussion

### 2.1. Assessing Therapeutic Efficacy and Safety of OIDs—Current Preclinical Testing Strategy

Preclinical studies of new OID candidates generally start from compound profiling in high-throughput in vitro studies [80]. Compounds with promising efficacy results progress to in vivo studies. Three preclinical animal-based studies are currently required by regulatory authorities before approving the request of clinical study for a novel OID. These are: (i) the range finding study, (ii) the repeat dose study, and (iii) the carcinogenicity study. Other specialized studies can be necessary, such as safety pharmacology studies, reproductive studies, and neonatal/juvenile studies for pediatric OIDs. Animal-based inhalation studies are carried out mainly in rats, mice or rabbits by exposure in restraint tubes [81]. Dogs and primates can also be used for testing OIDs in more realistic settings, via facemasks or helmets [82].

#### 2.1.1. Limitations of Current Preclinical Inhalation Testing Strategy in Assessing Therapeutic Efficacy

Although high-throughput cell-based assays can provide insightful information at the early stages of preclinical development, the cell models used fall short in recapitulating the complex interactions between different cell types and tissues/organs occurring in human. Conventional in vitro models are in fact formed by one cell type grown as a flat, two-dimensional culture; thus, they are a simplistic representation of the human lung tissue [83]. Furthermore, many in vitro assays use transformed cell lines that exhibit gene and protein expression that strongly differ from their primary counterpart [83].

On the other hand, various uncertainties characterize the animal-based preclinical studies currently required for regulatory purposes. The first level of uncertainty is associated with the type of devices used to administer the OID to the animal. While clinical nebulizers can be used in the preclinical environment (upon small modifications), DPIs and pMDIs cannot be employed to expose animal models at the preclinical screening level, as these devices are breath actuated. To overcome this issue, specialized equipment is used to expose the animal to an aerosol in a restrained environment. Aerosol of powders is achieved via, for example, rotating brush generators or Wright dust feed. An algorithm-based extrapolation [84] is then applied to define dose ranges to be used in clinical trials. The delivered dose is calculated as the amount of OID per unit of body weight that is presented to the animal. Due to the two parameters (velocity and aerodynamic size distribution) affecting OID deposition patterns in the lungs, as discussed in the section above, and to the species of the animal model used, the deposited dose is only a fraction of the delivered dose. The FDA assumes 100% deposition in humans, 10% in rats and 25% in dogs or non-human primates, irrespective of any information that has been produced by the submitting company [85]. This indeed generates uncertainties when calculating clinical overages.

The second level of uncertainty in in vivo studies is posed by the animal model itself [86]. For example, rodents are obligate nose breathers; this strongly influences how inhaled compounds deposit in the respiratory tract. This and other interspecies differences have been extensively discussed by the authors in a recent perspective [59]. Preclinical studies during OID development requires a clear understanding of such interspecies differences and their impact on the screening outcomes in terms of OID efficacy, toxicity and recovery from adverse effects.

Although not required at regulatory level, disease animal models are also used in preclinical research, particularly in the oncological field, as proof of concept for demonstrating OID efficacy. The authors have performed a literature search on PubMed using the searching terms “(inhaled drug) AND (in vivo) AND (efficacy)”. The search results showed that, in the last five years, 116 articles used disease animal models to test the efficacy of OIDs. However, animal use as disease models needs to be viewed cautiously. In animal models, disease features are reproduced by applying exogeneous stimuli (e.g., allergens, irritant gas exposures, cigarette smoke, etc.) [87]. This modelling process is however incomplete, as the use of single stimuli does not mimic the disease etiology and chronicity observed in patients.

The next section of this commentary focuses on this specific aspect, complementing the authors’ previous publication [59] and further discussing if and how new approach methodologies (NAMs) could become useful in the attempt to overcome the limitations of current animal models and increase OID translation rate. For completeness, it should be mentioned here that the abbreviation “NAMs” is often used in toxicology to refer broadly to any non-animal technology, methodology, approach, or combination thereof that can be used to provide information on chemical hazard and risk assessment. Examples of NAMs include non-mammalian model systems, (e.g., *Caenorhabditis elegans* [88,89,90], *Drosophila melanogaster* [91,92,93], zebrafish [94,95,96] and *Dictyostelium* [97]) and computational (in silico) approaches [98], which indeed offer opportunities for mimicking human respiratory diseases in a predictive manner. However, the scope of the NAMs considered in our commentary includes only in vitro, non-animal cell models for the testing of OIDs.

### 2.2. In Vitro Cell-Based NAMs for OID Efficacy Testing

Based on the most recent advances in tissue-engineering technologies, in vitro cell-based NAMs for screening the efficacy of OIDs can be classified in three main categories [99]: (i) tissue-mimetic lung cultures grown at the Air–Liquid Interface (ALI); (ii) lung organoids; and (iii) lung-on-chip.

#### 2.2.1. ALI Cultures

ALI cultures mimic one of the main properties of the lung epithelium, i.e., the direct contact with the gas phase (air). This provides a tissue-mimetic environment that makes it possible for airway epithelial cells to proliferate and differentiate in vitro into a pseudostratified, ciliated epithelium that produces mucus. Thus, ALI cultures provide an excellent method for testing OID dissolution and absorption, while enabling testing of the drug in its aerosol form. Whitcutt et al., were among the first research groups to report mucociliary differentiation in ALI cultures [100]. Today, ALI cultures are known to be particularly useful in understanding the mechanisms of respiratory diseases, including the cell-cell and cell-extracellular matrix interactions during airways remodeling [101,102,103]. Also, they can replicate some of the key features that need to be kept into account when developing an inhalation therapy, namely (i) the constitution and thickness of the pulmonary lining fluid [67] and (ii) mucociliary clearance [60,61,62]. For example, ALI cultures have been used to model the effects of smoke exposure on epithelial cells [104] and the authors have created a complex, diseased ALI culture model capable of reproducing the chemoresistance mechanisms observed in patients affected by non-small-cell lung cancer [105,106]. Also, culturing human airway epithelial cells isolated from patients, makes it possible to conduct patient-specific research and drug-screening, for example in cystic fibrosis, asthma and COPD [107,108,109,110]. With the aim of further increasing the predictive value of this in vitro NAM, ALI co-cultures have also been developed. In ALI co-cultures, the lung cell populations are mixed or partially separated, depending on the experimental set-up. In general, the immune cells are cultured in direct contact with the epithelial cells; whereas, fibroblasts and endothelial cells are separated from the epithelial cells by the transwell permeable membrane. Cell separation is due to the relative difference in the culturing conditions of the various cell types and the consequent need to separate them. This constitutes one of the main limitations of ALI models, as separated cells cannot establish physical (cell-to-cell) interactions as per in vivo conditions. This indeed affects the detected responses during OID preclinical testing.

#### 2.2.2. Lung Organoids

The second type of in vitro, cell-based NAMs currently available for OID testing are lung organoids. These are grown from human induced pluripotent stem cells (iPSCs) cultured within a natural or synthetic extracellular matrix to form three-dimensional (3D), hollow cell spheroids of basal, ciliated and secretory cells [111]. Through differentiation and self-organization of the iPSCs, an in vitro culture with lung tissue-specific morphogenetic and histological properties is formed [112]. To date, several organoids representative of the various human lung regions [39] and assessing a variety of pulmonary diseases [39,113,114] have been developed. In the context of OID preclinical testing, lung organoids can be used for modeling respiratory diseases and, therefore, as a platform for screening the efficacy of inhalation therapies [115,116]. Indeed, technical limitations are inherent with the use of lung organoids. Lungs are in fact subjected to mechanical deformation during breathing cycles, a deformation that is currently hard to model in organoids. Furthermore, there is still a lack of established in vitro lung organoids with a functional representation of the vasculature network. Most importantly, lung organoids lack an important feature for OID testing, i.e., the direct contact of epithelial cells with the air. As mentioned above, lung organoids are spherical cultures. They present an interiorized lumen, with epithelial cells facing inwards rather than outwards; this makes drug administration extremely difficult and reduces the application of organoids in the screening of OID absorption.

#### 2.2.3. Lung-on-Chip

Microfluidic technologies allow to add further complexity and functionality to the in vitro ALI models described above. The so-called “lung-on-chip” is a microfluidic-based in vitro system in which lung epithelial cells are grown on one side of a membrane, and stromal cells on the other surface. Liquid and air are circulated through the system to mimic air and blood flow in the lung. The applications of lung-on-chip range from basic research to drug discovery [117], where the OID can be introduced in the air flow as per in vivo conditions. Probably the most famous example of this in vitro, cell-based NAM is the breathing lung-on-chip developed by Huh and co-workers at the Wyss Institute of Harvard University (USA), capable of reproducing both the physiological and pathological responses of the human lung, a rudimentary circulatory system and the mechanical stress associated with breathing [118,119,120]. The immediate application of lung-on-chip has been for toxicity testing [121,122]; more recently, this model has been exploited for improving understanding of the complex lung disease processes and their responses to therapeutics [123,124,125], with applications extending even to the most recent need of a fast drug discovery for COVID-19 treatment [126]. Lung-on-chip systems allow, in fact, the in vitro creation of highly tissue-mimetic lung disease models [127,128], thus allowing, for example, to model the human response and the effects of existing and novel therapeutics when the lung is infected by the influenza virus or by viral pseudoparticles expressing spike protein of SARS-CoV-2, the virus responsible for COVID-19 development [126].

The clear advantage of lung-on-chip systems over ALI cultures or lung organoids is the possibility of mimicking the pulmonary mechanical stretch during in- and exhalation, while replicating the air-blood barrier for studying OID absorption. Furthermore, lung-on-chip models allow evaluating the impact of the mucociliary clearance mechanism overcoming the lack of directionality in cilia beating function characteristic of fully-differentiated in vitro ALI models [63]. Nevertheless, the lung-on-chip models share some of the limitations of ALI cultures, i.e., the impairment of physical crosstalk among different cell types. In fact, even in the most recent and advanced developments in “tumor-on-a-chip” cell culture technology, successfully used to create in vitro human orthotopic models of non-small-cell lung cancer [129], the lung cancer cells (cultured under ALI conditions) are physically separated from the lung endothelial cells by a porous, permeable membrane [130].

#### 2.2.4. In Vitro Cell-Based NAMs with Future Potential Application in OID Development and Testing

It is noteworthy to mention that, in the respiratory disease field, two additional categories of in vitro, cell-based NAMs exists, although these have not been used for OID testing to date. The first category is constituted by explant or ex vivo cultures, namely isolated perfused lungs and precision cut lung slices. These are better representations of the in vivo situation than any of the previous three NAM types mentioned above. The use of ex vivo cultures in OID testing is however hindered by the hurdles associated with their manipulation, and by donor-specific differences that make the OID screening outcomes often not significant or difficult to interpret [131]. The second category includes the engineered, reconstructed lung organs [132]. These are formed from several cell types co-cultured within scaffolds that aim at replicating the composition and architecture of the human lung acellular stroma [133]. Mechanical or biochemical stimuli can be added to tailor the properties of the scaffold and increase the similarity to the lung stroma in vivo. The first engineered lung organ was built from a decellularized lung matrix used as scaffold [134]. More recently, 3D bioprinting techniques have been used to produce the lung organs in vitro. For 3D bioprinting, cells are combined with bioactive hydrogels composed of synthetic (e.g., polyethylene glycol, pluronic) or natural (collagen, chitosan, fibrin, gelatin, Matrigel, alginate) polymers [135]. The use of reconstructed lung organs in OID preclinical screening is currently hampered by the low throughput of these methods.

## 3. Conclusions

To summarize, in this commentary we have presented an overview of the in vitro, cell-based NAM systems that, to date, have been successfully employed to fill the technological gap that is believed to hindering the effective OID translation from the lab bench to the clinic. In the past, OID failure at clinical trial stage was mainly due to poor pharmacokinetics and bioavailability. Today, these are rarely a cause of failure, as the pharmaceutical industry greatly invested in the development and application of much more accurate prediction and modelling approaches. Lack of efficacy is now the most common cause of OID attrition [11]; this appears to be associated to the fact that preclinical animal models are poorly representative of human respiratory diseases [136]. Improved in vitro non-animal methods could provide a more human-relevant predictive value so that compounds would fail earlier in their course of development [137]. Furthermore, we have provided a brief overview of those in vitro, cell-based NAMs that, in the future, we believe they could be adapted towards OID testing.

Although in vitro, cell-based NAMs still have limitations, the advantages associated with their use is evident and future efforts should aim at validating these systems for regulatory acceptance [59]. In the development of OIDs, we should therefore invest in moving away from animal studies. In the last decades, significant funding and precious time have been spent on developing animal models, despite the known species differences that make the results obtained from such models often unreliable when translated to humans. As Dr. Francois Busquet and colleagues from the Center for Alternatives to Animal Testing-Europe state for COVID-19, human-relevant approaches offer crucial advantages of speed and “much more robust and exacting data than any animal experiment could deliver” [138]. In this instance, we believe it is important to highlight that Directive 2010/63/EU on the protection of animals used for scientific purposes aims non only at reducing but at the “full replacement of procedures on live animals for scientific and educational purposes, as soon as it is scientifically possible to do so” [139]. Consistently with this aim, in 2016 The Netherlands has been the first EU member state to present a roadmap for phasing out animal testing in the safety research on chemical substances, food ingredients, pesticides and medicines (including veterinary medicines) [140]. The recent advances in tissue engineering, microfluidic and organ-on-chip technologies are providing researchers with tools for the development of human-relevant, in vitro NAMs. Thus, it is essential now that the respiratory disease research community embraces these tools, bringing them forward towards regulatory validation.

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
