# Peer review of "Preclinical Development of Orally Inhaled Drugs (OIDs)—Are Animal Models Predictive or Shall We Move Towards In Vitro Non-Animal Models?"

_animals, 2020, doi:10.3390/ani10081259_

Round 1
Reviewer 1 Report
This is a thorough and well-written comments on problems with transferring data from preclinical research to clinical products, and the authors show that there are challenges in this field, inhalation drugs. In this way, the manuscript contributes to the overall discussion of the preclinical tranlational crisis in which the research on experimental animals is today. It is interesting that the authors have chosen to contribute in both a very relevant field (orally inhaled drugs) and with specific suggestions on these issues. The authors should also be commended for thoroughly introducing the issues.
Major comments
- My main comment is that the conclusion of the manuscript is that one should bet on the in vitro models. Is there no need to improve both in vivo and in vitro models? The fact that in vitro models in some cases work very well does not necessarily mean that they can replace all in vivo studies.
- Can examples of test drugs be provided that contain both in vitro, in vivo and clinical data? It would be interesting to see how such data could be added to the text.
- If the format permits this, reading the text through subheadings will facilitate readability.
Minor comment
Line 61: Inhalation is the preferred administration method for treating respiratory diseases. Is that really true for all the diseases listed in the introduction? If not, this must be specified.
Author Response
Reviewer’s comment #1: This is a thorough and well-written comments on problems with transferring data from preclinical research to clinical products, and the authors show that there are challenges in this field, inhalation drugs. In this way, the manuscript contributes to the overall discussion of the preclinical translational crisis in which the research on experimental animals is today. It is interesting that the authors have chosen to contribute in both a very relevant field (orally inhaled drugs) and with specific suggestions on these issues. The authors should also be commended for thoroughly introducing the issues.
Authors’ response: We really thank the Reviewer for his/her positive feedback on our manuscript.
Reviewer’s comment #2: My main comment is that the conclusion of the manuscript is that one should bet on the in vitro models. Is there no need to improve both in vivo and in vitro models? The fact that in vitro models in some cases work very well does not necessarily mean that they can replace all in vivo studies.
Authors’ response: The take-home message of our manuscript is a call for a more human-relevant research in OID development and testing. Indeed, it is the authors’ opinion that, to achieve this, we should invest in moving away from animal studies. In the last decades significant funding and precious time has already been spent on developing animal models, despite the known interspecies differences that make the results obtained from such models often unreliable when translated to humans. As Dr Francois Busquet and colleagues from the Center for Alternatives to Animal Testing-Europe state for COVID-19, yet another respiratory disease, human-relevant approaches offer crucial advantages of speed and “much more robust and exacting data than any animal experiment could deliver”. In this instance, we believe it is important to highlight that Directive 2010/63/EU on the protection of animals used for scientific purposes aims not only at reducing but at the “full replacement of procedures on live animals for scientific and educational purposes, as soon as it is scientifically possible to do so”. Consistently with this aim, in 2016 The Netherlands has been the first EU member state to present a roadmap for phasing out animal procedures in biomedical research. Thus, in the manuscript conclusions the authors call for the respiratory disease research community to embrace the new tools already available for focusing the efforts towards the development of human-relevant, in vitro, cell-based NAMs.
The above considerations have been added in the revised manuscript (page 10, line 454), to better support the conclusions of our commentary.
Reviewer’s comment #3: Can examples of test drugs be provided that contain both in vitro, in vivo and clinical data? It would be interesting to see how such data could be added to the text.
Authors’ response: The authors agree with the Reviewer on this comment - A study of this kind would indeed be extremely interesting. Unfortunately, to the best of the authors’ knowledge following an extensive search through the scientific literature, studies that present the behaviour of an OID in vitro, in vivo as well as in the clinics are not available.
Reviewer’s comment #4: If the format permits this, reading the text through subheadings will facilitate readability.
Authors’ response: Following the Reviewer’s feedback, during the revision we have added additional subheadings in section 2.2.
Reviewer’s comment #5: Line 61: Inhalation is the preferred administration method for treating respiratory diseases. Is that really true for all the diseases listed in the introduction? If not, this must be specified.
Authors’ response: Inhalation is the preferred administration method for treating any respiratory disease, for the reasons described at page 2, line 63 and reported below for the Reviewers’ convenience:
“[...] (i) it delivers the drug directly at the site of action, resulting in a rapid therapeutic onset with considerably lower drug doses, (ii) it is painless and minimally invasive thus improving patients’ compliance, and (iii) it avoids first-pass metabolism, providing optimal pharmacokinetic conditions for drug absorption and reducing systemic side effects [13-15].”
References that back up this statement are also provided (references 13 to 15).
The respiratory diseases that currently can be treated with OIDs are asthma, COPD and cystic fibrosis, as indicated at page 2, line 77:
“OID therapeutic categories currently approved for the clinical treatment of respiratory diseases include drugs for the treatment of asthma and COPD, such as b2 adrenergic agonists (e.g. albuterol, formoterol) and muscarinic antagonists (e.g. ipratropium, tiotropium) inducing bronchodilation, or glucocorticosteroids (e.g. fluticasone and budesonide) reducing inflammation. OIDs for the treatment of cystic fibrosis are also available for clinical use, with most of them falling into the therapeutic categories of mucolytics (e.g. saline and acetyl choline), aiming at thinning the mucus for facilitating its clearance from the patient’s lungs. Alternatively, leukocyte DNAse, reducing inflammation, and antimicrobial agents (e.g. tobramycin), treating the bacterial infection characteristic of this disease, are also administered as OIDs.”
Reviewer 2 Report
My overarching feedback is that I think you are trying to do too much with this paper and it would benefit from more focus in some areas. I think that the two main ideas that I took from the paper- the drivers for drug development for inhaled delivery and non-animal methods to reduce animal testing - are not linked and so the paper feels slightly disjointed to me. You present many convincing facts but at times the arguments are not quite finished and I was left wondering how and why some of the sections are relevant to your title.
You spend some time describing devices and whilst I appreciate the value of this given the nature of the paper, it was my understanding that the devices themselves do not undergo animal testing and instead, may be used in other approaches (cascade impactors etc) to look at flow, deposition, particle size etc. I am not sure that I understand how the description of the devices or regulatory requirements (lines 61-90) are fully relevant in addressing the predictivity of animal models.
Line 99 - I am not sure that it is valuable, or addresses the title of the paper, to include the information about OID for systemic disease. This is interesting, but not particularly relevant.
Lines 112-117- the issues with drug deposition need to be addressed with alteration of delivery methods/drug formulation and this is not an issue associated with preclinical animal models per se, as animals are not involved in deposition/particle sizing studies. Some consideration of computation models of flow though the human airways, deposition modelling and simulation studies would be relevant here, but still these issues are not associated with animal use and addressing them should not impact animal use.
I do not agree with your use of the term 'barriers' in section 1.2.1. To me, particle size, deposition, anatomy, cellular composition are not barriers, but are all features that need to be taken into consideration when designing drugs for the airways, and therefore all can be surmounted. I feel that in order to make this paper a powerful set of reasons to move away from preclinical animal testing, the barriers are those that are currently driven by animal testing (lack of efficacy, species differences, disease progression, cellular complexity, etc) that can only be addressed by moving toward the in vitro methodologies that you are describing.
Lines 180-185 - very interesting but I am not sure of the relevance of lung metabolic activity, unless you can relate this to the values for the preclinical animal models, and use this as another reason to shift to human cell-based models.
Line 193 - I was pleased to see that you are using clinical trial data- but I do feel that this needs to be put into context. I have no idea whether 2,508 trials is a high number or very modest. For example, for the same time period, there were 2,733 trials for breast cancer alone and so over 2,000 trials for 115 diseases seems to indicate a relative lack of interest by Pharma. Also, how many of these interventional studies were behavioural and how many actually used potential new drugs?
Lines 206-207 - when you consider the use of animals, it might be useful to report the numbers of animals used in this research, again to put this into context and define the scale of the issue.
235-237 - This is more of a general comment as I totally agree with you here, but this issue goes beyond OID and is relevant for all preclinical testing. Interspecies differences are insurmountable. I am also not sure how much OID screening is done using disease models, I did not think that drug registration required efficacy testing, so presumably the disease models are not used for regulatory purposes? I can understand why you wish to include the information about disease models, but am not sure that it is appropriate here when there are two separate issues for me -one is preclinical testing for safety and the second is disease modelling. Both of which could be addressed with non-animal approaches, but they are very different.
Line 253- this is not unique to lung epithelium -the nature of epithelia mean that they all have an air interface.
Line 267 (and elsewhere) I would really love to see some figures in this paper, to consolidate some of the information and just help the reader visualise what you are discussing. I think that one figure could compare human and animal airways anatomy and cellular composition - to reinforce your point about species differences and also about drug deposition relative to particle size, and one figure could summarise the cell types employed in the non-animal approaches so far and their applications.
Line 284 you refer to the paper by Huh et al and I don't think that this is the most appropriate citation to support your statement. As I understood it, in Huh's paper, the lung chip was used to evaluate the possible therapeutic effects of a new experimental compound following stimulation of the cells with IL2 to induce oedema, and not for toxicity testing, as you seem to imply. The authors did conclude that their system could be useful for predicting the efficacies and toxicities of other drugs in humans - but I do not think that they show this in their study. My apologies if I misunderstand you here, but when I think of toxicity testing, I think of dose responses linked to adverse events and Huh at al did not show this - but I agree that the organ-chips are a valuable resource for toxicity testing and liver chips have been used to demonstrate this (Jang et al. "Reproducing Human and Cross-Species Drug Toxicities Using a Liver-Chip." Science Translational Medicine 11 (2019): eaax5516.).
I suggest a minor reorganisation of section 2.2 such that the most complex system (the lung chip) is considered last. To me, the sophistication of the chips accounts for some of the limitations that you describe for the organoids, and so it makes logical sense to me to describe how chips can overcome this limitation after you have detailed the issues with organoids.
Line 309: "Thus, organoids have no application in the screening of OID absorption" seems a very bold statement and I am not sure that I fully understand the rationale behind this. I appreciate that there are limitations to the organoids, but I disagree that they have no application at all and think that this statement should be qualified.
Lines 311-326 I am not sure why you are describing these other in vitro models if these have not been used for OID testing, unless you can tell the reader how and why they might be utilised for this purpose in the future and how this would contribute to a reduction in animal use. I think it is valuable to indicate the breadth of human-relevant approaches that exist or are under development, but this would be more powerful if you could suggest how these might be used- in combination with other methods perhaps or for more accurate disease modelling?
I did feel that the discussion of the non-animal methods was not linked to the initial section about drug deposition and particle size. I wonder if there is scope for you to consider the non-animal methods more widely, going beyond in vitro to consider active projects that are using in silico methods, 3D computer modelling of the respiratory tract and computational fluid dynamics modelling of airflow. Whilst these methods may not directly replace the animals used in toxicity testing, they do address some of the issues that you raise regarding particle-size dependent drug deposition.
I also felt that the comment regarding attrition and lack of efficacy (line 336) is not fully addressed throughout the rest of the paper and it is not clear to me how the application of the non-animal methods would address this - do the authors have any examples that they could use to support this? There are a couple of references provided, but these are for reviews rather than more precise examples of the application of non-animal methods toward animal replacement for drug efficacy testing. The example of Si et al is probably the most relevant and maybe the authors could expand on how this was used and what the future promise could be?
My last point is very pedantic, but I think it might be useful to share. I understand NAM to mean new approach methodologies- as described by ICCVAM in 2018 - this includes in silico methods, not just in vitro methods and some people also include some animal-based models as NAMs. This acronym is widely used across the US at least to refer to new approach methodologies and I found it very confusing for you to repurpose NAM as non-animal methods.
Author Response
Reviewer’s comment #1: My overarching feedback is that I think you are trying to do too much with this paper and it would benefit from more focus in some areas. I think that the two main ideas that I took from the paper the drivers for drug development for inhaled delivery and non-animal methods to reduce animal testing - are not linked and so the paper feels slightly disjointed to me. You present many convincing facts but at times the arguments are not quite finished and I was left wondering how and why some of the sections are relevant to your title. You spend some time describing devices and whilst I appreciate the value of this given the nature of the paper, it was my understanding that the devices themselves do not undergo animal testing and instead, may be used in other approaches (cascade impactors etc) to look at flow, deposition, particle size etc. I am not sure that I understand how the description of the devices or regulatory requirements (lines 61-90) are fully relevant in addressing the predictivity of animal models.
Authors’ response: We thank the Reviewer for his/her insight, based on which we have completely revised the manuscript structure and content, deleting information that might be less relevant and focusing on those aspect that support our call to move towards more human-relevant models in OID testing. We have also focused our efforts on connecting the various sections through logical links, so to increase the readability of the manuscript.
As first main point to be addressed, we have revised the section on the inhalation devices, as indicated by the Reviewer in the comment above, to underline that these undergo characterization via in vitro, ex vivo and in vivo (human) testing, while animal models are not used in this instance (page 3, line 97). The description of the regulatory guidelines within the same paragraph has also been shortened, as suggested by the Reviewer.
Reviewer’s comment #2: Line 99 - I am not sure that it is valuable, or addresses the title of the paper, to include the information about OID for systemic disease. This is interesting, but not particularly relevant.
Authors’ response: With the aim to highlight the full potential of inhalation therapy and the subsequent interest of the pharmaceutical industry for having at hand preclinical models that allow a quick translation of OIDs, this information has been retained. Nevertheless, the paragraph has been moved to page 2 (line 72) to increase the manuscript clarity.
Reviewer’s comment #3: Lines 112-117- the issues with drug deposition need to be addressed with alteration of delivery methods/drug formulation and this is not an issue associated with preclinical animal models per se, as animals are not involved in deposition/particle sizing studies. Some consideration of computation models of flow though the human airways, deposition modelling and simulation studies would be relevant here, but still these issues are not associated with animal use and addressing them should not impact animal use.
Authors’ response: The Reviewer is indeed correct in his/her remark, as animal studies are not used for determining OID deposition pattern. Thus, the following sentence has been added at page 4, line 184:
“OID deposition pattern is currently evaluated in in vitro, cell-free experiments, achieving good predictive value [50].”
The reference cited here refers to a recent review describing in detail the computational (in silico) models that permit improved prediction of extrathoracic and lung deposition fractions in a variety of age groups, breathing conditions and variable conditions in both size and shape of the upper airway.
Reviewer’s comment #4: I do not agree with your use of the term 'barriers' in section 1.2.1. To me, particle size, deposition, anatomy, cellular composition are not barriers, but are all features that need to be taken into consideration when designing drugs for the airways, and therefore all can be surmounted. I feel that in order to make this paper a powerful set of reasons to move away from preclinical animal testing, the barriers are those that are currently driven by animal testing (lack of efficacy, species differences, disease progression, cellular complexity, etc) that can only be addressed by moving toward the in vitro methodologies that you are describing.
Authors’ response: During the revision of section 1.2.1, the authors have taken into great account the Reviewers’ comment above, and changed the term “barriers” into “features”, as suggested. Furthermore, we have clearly highlighted which of these features could be mimicked by adopting in vitro, cell-based NAMs, thus producing a clear benefit towards the reduction of the current high attrition rate of OIDs and supporting the manuscript call for the uptake of non-animal in vitro models in the OID development pipeline. The added information is highlighted in track-changes mode at page 4-6.
Reviewer’s comment #5: Lines 180-185 - very interesting but I am not sure of the relevance of lung metabolic activity, unless you can relate this to the values for the preclinical animal models, and use this as another reason to shift to human cell-based models.
Authors’ response: The following statement has been added to shed some light on this aspect (page 5, line 233): “Animal models and humans differ in the metabolism and distribution/types of cell populations lining the airways. [...] The human-specific composition and metabolism of the lung can indeed be replicated more closely by adopting in vitro, cell-based NAMs, as described in the following sections.”
Reviewer’s comment #6: Line 193 - I was pleased to see that you are using clinical trial data- but I do feel that this needs to be put into context. I have no idea whether 2,508 trials is a high number or very modest. For example, for the same time period, there were 2,733 trials for breast cancer alone and so over 2,000 trials for 115 diseases seems to indicate a relative lack of interest by Pharma. Also, how many of these interventional studies were behavioural and how many actually used potential new drugs?
Authors’ response: Clinical trial data reported in the manuscript were rechecked at the time of revision (using the same search terms) and corrected as per current date. As indicated by the search terms listed at page 5 (line 248-249), the 2,508 trials found on ClinicalTrial.Gov are all interventional studies. Unfortunately, identifying which of these trials tested new drugs requires a metadata analysis outside the scope of the current manuscript. Nevertheless, we have indicated the total drug interventions that were tested (666). Furthermore, to put the data into context, we have included the total number of interventional clinical trials registered in the same time period, showing that inhalation clinical trials make for the 2.6% of the total number of interventional studies registered in the time period under consideration (2016-2020), and 23.2% of total drug interventions examined (page 5, line 249-253).
Reviewer’s comment #7: Lines 206-207 - when you consider the use of animals, it might be useful to report the numbers of animals used in this research, again to put this into context and define the scale of the issue.
Authors’ response: During the manuscript preparation the authors looked for the number of animals used for OID development, using as reference the most recent report available in the field, which was published in 2019 by the Commission of the European Parliament and the Council and reports the statistics on the use of animals for scientific purposes in the Member States of the European Union in the 2015-2017 period (https://eur-lex.europa.eu/legal-content/EN/TXT/PDF/?uri=CELEX:52020DC0016&from=EN). Unfortunately, breakdowns for specific drug categories are not provided. A systematic review of the published scientific literature on the topic would be an interest step to follow up from the present manuscript to address this point. We would like to thank the reviewer for the suggestions.
Reviewer’s comment #8: Lines 235-237 - This is more of a general comment as I totally agree with you here, but this issue goes beyond OID and is relevant for all preclinical testing. Interspecies differences are insurmountable. I am also not sure how much OID screening is done using disease models, I did not think that drug registration required efficacy testing, so presumably the disease models are not used for regulatory purposes? I can understand why you wish to include the information about disease models, but am not sure that it is appropriate here when there are two separate issues for me - one is preclinical testing for safety and the second is disease modelling. Both of which could be addressed with non-animal approaches, but they are very different.
Authors’ response: As stated at page 6, line 260: “Three preclinical animal-based studies are currently required by regulatory authorities before approving the request of clinical study for a novel OID. These are (i) the range finding study, (ii) the repeat dose study, and (iii) the carcinogenicity study.” The Reviewer is indeed correct in his/her comment, as these studies use healthy animal models. Although not required at regulatory level, however, disease animal models are also used in OID preclinical research, particularly in the oncological field, as proof of concept for demonstrating the drug efficacy. The authors have performed a literature search on Pubmed using the searching terms “(inhaled drug) AND (in vivo) AND (efficacy)”. The search results showed that, in the last five years, 116 articles used disease animal models to test the efficacy of OIDs. This information has been added at page 6 (line 296) of the revised manuscript. We believe this has improved the quality of the information provided within our manuscript in line with the Reviewer’s comments.
Reviewer’s comment #9: Line 253- this is not unique to lung epithelium -the nature of epithelia mean that they all have an air interface.
Authors’ response: To avoid misunderstandings, the term “unique” has been removed from the sentence. The revised sentence reads as follows (page 7, line 324):
“ALI cultures mimic one of the main properties of the lung epithelium, i.e. the direct contact with the gas phase (air).”
Reviewer’s comment #10: Line 267 (and elsewhere) I would really love to see some figures in this paper, to consolidate some of the information and just help the reader visualise what you are discussing. I think that one figure could compare human and animal airways anatomy and cellular composition – to reinforce your point about species differences and also about drug deposition relative to particle size, and one figure could summarise the cell types employed in the non-animal approaches so far and their applications.
Authors’ response: We thank the Reviewer for this suggestion, which we have carefully considered. Figures similar to the ones described by the Reviewer are available in the literature; thus, the authors have decided to not include any figures (so to not duplicate the information widely available in literature). In line with the Reviewer’ previous comments, we believe this will also preserve the main take-home message, i.e. a call for moving away from animal studies.
Reviewer’s comment #11: Line 284 you refer to the paper by Huh et al and I don't think that this is the most appropriate citation to support your statement. As I understood it, in Huh's paper, the lung chip was used to evaluate the possible therapeutic effects of a new experimental compound following stimulation of the cells with IL-2 to induce oedema, and not for toxicity testing, as you seem to imply. The authors did conclude that their system could be useful for predicting the efficacies and toxicities of other drugs in humans - but I do not think that they show this in their study. My apologies if I misunderstand you here, but when I think of toxicity testing, I think of dose responses linked to adverse events and Huh at al did not show this - but I agree that the organ-chips are a valuable resource for toxicity testing and liver chips have been used to demonstrate this (Jang et al. "Reproducing Human and Cross-Species Drug Toxicities Using a Liver-Chip." Science Translational Medicine 11 (2019): eaax5516.).
Authors’ response: Thanks to the Reviewer’s comment, more appropriate references have been added to our statement on the application of lung-on-chip system for toxicity testing (references 124 and 125 of the revised manuscript). We thank the Reviewer for pointing out this mistake.
Reviewer’s comment #12: I suggest a minor reorganisation of section 2.2 such that the most complex system (the lung chip) is considered last. To me, the sophistication of the chips accounts for some of the limitations that you describe for the organoids, and so it makes logical sense to me to describe how chips can overcome this limitation after you have detailed the issues with organoids.
Authors’ response: The sections sequence has been revised as suggested by the Reviewer. The advantages of lung-on-chip systems over ALI cultures and lung organoids have also been clearly stated at page 9, line 403:
“The clear advantage of lung-on-chip systems over ALI cultures or lung organoids is the possibility of mimicking the pulmonary mechanical stretch during in- and exhalation, while replicating the air-blood barrier for studying OID absorption. Furthermore, lung-on-chip models allow evaluating the impact of the mucociliary clearance mechanism overcoming the lack of directionality in cilia beating function characteristic of fully-differentiated in vitro ALI models [115].”
We believe this revision has contributed to improving the manuscript.
Reviewer’s comment #13: Line 309: "Thus, organoids have no application in the screening of OID absorption" seems a very bold statement and I am not sure that I fully understand the rationale behind this. I appreciate that there are limitations to the organoids, but I disagree that they have no application at all and think that this statement should be qualified.
Authors’ response: We have carefully considered the Reviewer’s comment and revised the statement accordingly. We hope this now better reflects and explains the rationale behind our statement. The revised sentence (page 8, line 382) reads as follows:
“Most importantly, lung organoids lack an important feature for OID testing, i.e. the direct contact of epithelial cells with the air. As mentioned above, lung organoids are spherical cultures. They present an interiorized lumen, with epithelial cells facing inwards rather than outwards; this makes drug administration extremely difficult and reduces the application of organoids in the screening of OID absorption.”
Reviewer’s comment #14: Lines 311-326 I am not sure why you are describing these other in vitro models if these have not been used for OID testing, unless you can tell the reader how and why they might be utilised for this purpose in the future and how this would contribute to a reduction in animal use. I think it is valuable to indicate the breadth of human-relevant approaches that exist or are under development, but this would be more powerful if you could suggest how these might be used- in combination with other methods perhaps or for more accurate disease modelling?
Authors’ response: In this instance it is the authors’ opinion that, including a comprehensive description of all the in vitro, cell-based NAMs available to reproduce the lung, is informative. Our intention is that by listing these, we are reinforcing the message and call of moving towards a more predictive OID testing in the future.
Reviewer’s comment #15: I did feel that the discussion of the non-animal methods was not linked to the initial section about drug deposition and particle size. I wonder if there is scope for you to consider the non-animal methods more widely, going beyond in vitro to consider active projects that are using in silico methods, 3D computer modelling of the respiratory tract and computational fluid dynamics modelling of airflow. Whilst these methods may not directly replace the animals used in toxicity testing, they do address some of the issues that you raise regarding particle-size dependent drug deposition.
Authors’ response: We hope that this aspect has been address in the revision of the manuscript, as discussed in the authors’ response to the Reviewer’s comment #4.
Reviewer’s comment #16: I also felt that the comment regarding attrition and lack of efficacy (line 336) is not fully addressed throughout the rest of the paper and it is not clear to me how the application of the non-animal methods would address this - do the authors have any examples that they could use to support this? There are a couple of references provided, but these are for reviews rather than more precise examples of the application of non-animal methods toward animal replacement for drug efficacy testing. The example of Si et al is probably the most relevant and maybe the authors could expand on how this was used and what the future promise could be?
Authors’ response: Section 2.2 (now entitled “In vitro cell-based NAMs for OID efficacy testing”) addresses how the application of the non-animal methods can overcome the lack of OID efficacy seen in clinical trials. In the same section, we provide reference to studies where the in vitro, cell-based NAM systems have been used for OID efficacy testing. We report below the referenced paragraphs that, in our opinion, address the Reviewer’s comment.
1) ALI cultures - Page 7, line 335: “Also, culturing human airway epithelial cells isolated from patients, makes it possible to conduct patient-specific research and drug-screening, for example in cystic fibrosis, asthma and COPD [103-106].”
2) Lung organoids – Page 8, line 376 “In the context of OID preclinical testing, lung organoids can be used for modeling respiratory diseases and, therefore, as a platform for screening the efficacy of inhalation therapies [122,123].”
3) Lung-on-chip - Page 8, line 398: “more recently, this model has been exploited for improving understanding of the complex lung disease processes and their responses to therapeutics [111-113]”.
We hope this clarify our view by providing examples and additional references in line with the Reviewer’s request.
Reviewer’s comment #17: My last point is very pedantic, but I think it might be useful to share. I understand NAM to mean new approach methodologies- as described by ICCVAM in 2018 - this includes in silico methods, not just in vitro methods and some people also include some animal-based models as NAMs. This acronym is widely used across the US at least to refer to new approach methodologies and I found it very confusing for you to repurpose NAM as non-animal methods.
Authors’ response: We completely agree with the Reviewer on this point. A confusing use of the abbreviation “NAMs” is evident in the literature. The term NAMs is currently used as an acronym for both “new approach methodologies” and “non-animal methodologies”. This controversial use of the same abbreviation for two different concepts is demonstrated by two articles, both published in the same journal in 2020 (Punt et al., https://doi.org/10.14573/altex.2003242; Lorenzetti et al., https://doi.org/10.14573/altex.2003041). In EU, where the authors are based, the abbreviation “NAMs” is often used to indicate “non-animal methodologies”, as indicated in the European Consensus-Platform for Alternatives (Ecopa) Symposium. Nevertheless, to avoid any confusion, the authors have changed “non-animal methodologies” to “new approach methodologies” throughout the manuscript. Additionally, the scope of the NAMs described within our manuscript has now been clearly stated at page 6 (line 315):
“[...] the scope of the NAMs considered in our commentary includes only in vitro, non-animal cell models for the testing of OID efficacy.”
We hope this is providing the ground for the use of the term “NAMs” in our manuscript.